# Prediction of continuous amyloid positron emission tomography with fluid measures of phosphorylated tau and β-amyloid

Niklas Mattsson-Carlgren [1,2,3,15]✉, Linda Karlsson [1,15], Weizhong Tang[1], Kaj Blennow [4,5,6,7], Henrik Zetterberg [4,5,8,9,10,11], Randall J Bateman [12,13], Suzanne E Schindler[12,13], Nicolas Barthelemy[12,14], Sebastian Palmqvist [1,2], Erik Stomrud[1,2], Shorena Janelidze[1] & Oskar Hansson [1]✉

## Abstract

Brain amyloid-β (Aβ) pathology is a core feature of Alzheimer disease (AD) and can be quantified using positron emission tomography (PET). Cerebrospinal fluid (CSF) and plasma biomarkers detect abnormal Aβ, but it is unclear to what degree they can predict quantitative Aβ-PET. We explored plasma and CSF biomarkers in relation to Aβ-PET in the BioFINDER-2 study ($N = 1053$), and the BioFINDER-1 study ($N = 238$). We developed a machine learning pipeline to predict Aβ-PET using CSF and plasma measures. The best models achieved $R^2 = 0.79$. Plasma P-tau217 and CSF Aβ42/Aβ40 contributed the most. CSF Aβ42/Aβ40 contributed most to identify Aβ-positivity, while continuous Aβ-PET load within the positive range was best predicted by plasma P-tau217. Models using only plasma measures approached performance of CSF models. Altered metabolism of soluble Aβ may be highly associated with presence of Aβ plaques, while soluble P-tau217 levels may continue to change during build-up of Aβ pathology.

**Keywords** Alzheimer's Disease; Clinical Trials; Plasma Biomarkers; Amyloid; PET

**Subject Categories** Biomarkers; Neuroscience

## Introduction

Alzheimer's disease (AD) is the most common cause of dementia and is expected to continue to increase in global prevalence during the coming decades due to a growing elderly population. A key feature of AD is the accumulation of extracellular plaques, composed of β-amyloid (Aβ) peptides. Anti-Aβ-immunotherapies can both reduce the Aβ burden, and slow the progression of cognitive decline in AD patients (Sims et al, 2023; van Dyck et al, 2023). These novel treatments bring an increased need for accurate tools to identify and quantify Aβ pathology in living humans, first to support the AD diagnosis and make decisions about start of treatment, and second to monitor effects of treatment on the underlying pathology. For example, since some Aβ targeting therapy (e.g., donanemab) should be discontinued when the Aβ pathology has been sufficiently reduced, there is a need for scalable tools to quantify the amount of Aβ pathology after treatment.

Aβ aggregation can be detected and quantified in vivo with positron emission tomography (PET) (Clark et al, 2011), but this is expensive, has low availability, and is difficult to use on a large scale (especially in a global context). As an alternative, Aβ-pathology can also be detected by fluid biomarkers, including both cerebrospinal fluid (CSF) (Mattsson-Carlgren et al, 2022) and plasma biomarkers (Hansson, 2021; Hansson et al, 2023). Key biomarkers that are altered in the presence of Aβ-pathology, and that have been examined in relation to Aβ-PET, include measures related to Aβ-metabolism (Aβ42/Aβ40 (Schindler et al, 2019; Brand et al, 2022)), abnormal phosphorylation of tau (e.g., P-tau181 (Janelidze et al, 2020; Kwon et al, 2023), P-tau217 (Palmqvist et al, 2020; Milà-Alomà et al, 2022; Mattsson-Carlgren et al, 2021; Li et al, 2024) and P-tau231 (Milà-Alomà et al, 2022; Ashton et al, 2021)), and astrogliosis (e.g., glial acidic fibrillary protein, GFAP)(Pereira et al,

[1]Clinical Memory Research Unit, Department of Clinical Sciences Malmö, Lund University, Lund, Sweden. [2]Memory Clinic, Skåne University Hospital, Malmö, Sweden. [3]Wallenberg Center for Molecular Medicine, Lund University, Lund, Sweden. [4]Department of Psychiatry and Neurochemistry, Institute of Neuroscience and Physiology, the Sahlgrenska Academy at the University of Gothenburg, Mölndal, Sweden. [5]Clinical Neurochemistry Lab, Sahlgrenska University Hospital, Mölndal, Sweden. [6]Paris Brain Institute, ICM, Pitié-Salpêtrière Hospital, Sorbonne University, Paris, France. [7]Neurodegenerative Disorder Research Center, Division of Life Sciences and Medicine, and Department of Neurology, Institute on Aging and Brain Disorders, University of Science and Technology of China and First Affiliated Hospital of USTC, Hefei, P. R. China. [8]Wisconsin Alzheimer's Disease Research Center, University of Wisconsin School of Medicine and Public Health, University of Wisconsin-Madison, Madison, WI, USA. [9]Department of Neurodegenerative Disease, UCL Institute of Neurology, London, UK. [10]UK Dementia Research Institute at UCL, London, UK. [11]Hong Kong Center for Neurodegenerative Diseases, Clear Water Bay, Hong Kong, China. [12]Department of Neurology, Washington University School of Medicine, Saint Louis, MO, USA. [13]Charles F. and Joanne Knight Alzheimer Disease Research Center, Washington University School of Medicine, St. Louis, MO, USA. [14]Tracy Family SILQ Center for Neurodegenerative Biology, St. Louis, MO, USA. [15]These authors contributed equally: Niklas Mattsson-Carlgren, Linda Karlsson.✉E-mail: Niklas.mattsson-carlgren@med.lu.se; oskar.hansson@med.lu.se

2021). Fluid measures of neurofilament light (NFL) (Mattsson et al, 2017) and GFAP (Fenoglio et al, 2024) are also altered in AD, but are more non-specific indicators of neurodegenerative pathology.

Many studies have evaluated fluid biomarkers for their ability to differentiate Aβ-positivity versus negativity, defining Aβ-status by PET or (for plasma biomarkers) CSF Aβ measures. Importantly from a diagnostic standpoint, the distribution of CSF Aβ42/Aβ40 values is distinctly bimodal even in an unselected patient cohort, and there is a very high concordance with Aβ PET for patients with low CSF Aβ42/Aβ40 values (Gobom et al, 2022). A meta-analysis also showed very high agreement between CSF Aβ42/Aβ40 assayed using different technologies and Aβ PET, with AUC of 0.96 for large consecutive memory clinic cohorts (Shaw et al, 2018). The best fluid biomarkers have high performance for this classification task, across a range of thresholds for Aβ-positivity (Palmqvist et al, 2020; Barthélemy et al, 2024). However, results for plasma models developed with CSF as reference standard may not translate directly to prediction of Aβ PET (Janelidze et al, 2024).

Moving beyond classification, studies that have assessed continuous correlations between fluid biomarkers levels and Aβ-PET burden have typically reported low to moderate correlations for Aβ fluid measures (Brand et al, 2022; Mundada et al, 2023; Janelidze et al, 2020; Ashton et al, 2021) and moderate to high correlations for P-tau measures (Mundada et al, 2023; Janelidze et al, 2020; Ashton et al, 2021), when including the full range of Aβ PET-uptake. When studies have stratified individuals by Aβ-status, correlations have been weak or absent within the Aβ-negative range, and attenuated (Janelidze et al, 2020; Wisch et al, 2023) or even absent (Mundada et al, 2023) within the Aβ-positive range (Barthélemy et al, 2023). The correlations with continuous Aβ-PET load within the Aβ-PET-positive range could also differ for CSF and plasma measures, even for the same analyte, e.g., Aβ42/Aβ40 (Wisch et al, 2023) (where CSF was more strongly associated with Aβ-PET load than plasma). Most studies of fluid biomarkers predicting continuous Aβ-PET have not rigorously evaluated model performance out-of-sample and lack large enough samples sizes to perform robust out-of-sample evaluations. These studies may therefore lack evidence on model generalizability, which is of uttermost importance to draw conclusion about a biomarker's predictive ability.

Taken together, while CSF Aβ42/Aβ40 (Shaw et al, 2018) and CSF P-tau181/Aβ42 (Hansson et al, 2018) can predict brain Aβ-PET positivity with very high accuracy (identical to the between-rater agreement for visual Aβ-PET classification by expert readers), it is unclear to what degree fluid biomarkers can measure the actual Aβ-burden as quantified by PET, something that is of very high relevance in the current era of Aβ targeting therapies. We set out to develop an efficient machine learning pipeline that could harness the potential of a broad panel of fluid AD biomarkers, including Aβ42/Aβ40, different P-tau variants, GFAP and NFL, to predict Aβ-PET standardized uptake volume ratio (SUVR). We aimed to identify the best biomarker model, and the most important biomarkers for Aβ-PET predictions. We also aimed to clarify if different classes of fluid biomarkers (e.g., Aβ-related versus tau-related) were mainly associated to Aβ-PET status (i.e., negative versus positive) or the continuous Aβ-PET burden (specifically the load within the positive range). We focused mainly on plasma biomarkers, but we also incorporated CSF biomarkers, both to provide an upper benchmark for potential performance of fluid biomarkers, and to efficiently meet the aim related to the role of different classes of fluid biomarkers. Our a priori hypothesis was that P-tau217 and Aβ42/Aβ40 measures would have the strongest association with continuous Aβ-PET, considering that they have been robustly associated with Aβ in the past (Schindler et al, 2019; Mattsson-Carlgren et al, 2021; Wisch et al, 2023; Ashton et al, 2022), and that their effects have been shown to be complimentary both when predicting Aβ-status in vivo (Janelidze et al, 2022), and post-mortem Aβ-pathology (Salvadó et al, 2023). For comprehensiveness, we also evaluated the potential contributions of basic demographic variables and cognitive test scores to the models.

# Results

The first part of the study included a total of 1140 participants (51.8% female, mean age 66.5) from the BioFINDER-2 study, focusing on participants with plasma P-tau217 data (measured with mass spectrometry [MS], BF2-P-MS, n = 1053). Most of the participants were cognitively unimpaired (CU, 50.8% normal controls, 19.0% subjective cognitive decline [SCD]). Among the cognitively impaired (CI), most had mild cognitive impairment (MCI, 28.5% of all) and only a few had dementia (1.7%). The overall rate of Aβ-PET positivity was 37.1% (consistent with that the cognitively unimpaired study arms of BioFINDER-2 were enriched for APOE ε4 carriers and with established global prevalence estimates (Jansen et al, 2022)). Different subsets were analyzed in different parts of the workflow, depending on data availability for different fluid biomarkers, with counts ranging from 254 to 1053 individuals (Tables 1 and EV1). For external validation, we included 238 participants from BioFINDER-1 (Table 2).

## Model development and feature engineering

We established an initial sub-cohort, BF2-Initial (Table 1), which focused on a wide range of plasma biomarkers (the sub-cohort was restricted to those with all data available), including plasma Aβ42/Aβ40, P-tau231, P-tau217$_{WU}$ (WU, Washington University), P-tau217$_{Li}$ (Li, Lilly), P-tau205, P-tau181, %P-tau217$_{WU}$, %P-tau181, %P-tau205, NFL, NTA, and GFAP, as well as the three most available CSF measures: CSF P-tau217, CSF P-tau181, and CSF Aβ42/Aβ40. BF2-Initial was used for model development and feature engineering (Appendix Table S1).

## Initial model selection

We performed an initial training of 10 different machine learning regressors with proper tuning (Appendix Table S2), to identify the model(s) with best overall potential for prediction of quantitative Aβ-PET. Seven tree-based and boosting models had overall similar cross-validation performance ($R^2$ 0.774–0.827, highest for Extra Trees Regressor), while the three other models (Ridge regression, SVM, and K-neighbors) had lower overall performance ($R^2$ 0.462–0.695) (Appendix Table S3). Based on its overall high performance, we proceeded with the Extra Trees Regressor for the main experiments. We also performed sensitivity analyses with the Gradient Boosting Regressor, which was one of the best boosting models tested.

**Table 1. Demographics of participants from the Swedish BioFINDER-2 Study.**

| | | BF2 | BF2-Initial | BF2-P-MS |
|---|---|---|---|---|
| n | | 1140 | 499 | 1053 |
| Age, y [mean (sd)] | | 66.5 (13.3) | 65.9 (13.7) | 66.3 (13.5) |
| Sex [n (%)] | Male | 549 (48.1) | 244 (48.9) | 510(48.4) |
| | Female | 591 (51.8) | 255 (51.1) | 543 (51.6) |
| Education, y [mean (sd)] | | 12.9 (3.69) | 12.8 (3.64) | 12.9 (3.70) |
| Cognitive status [n (%)] | CN | 579 (50.8) | 263 (52.7) | 548 (52.0) |
| | SCD | 217 (19.0) | 93 (18.6) | 195 (18.5) |
| | MCI | 325 (28.5) | 141 (28.3) | 292 (27.7) |
| | Dementia[a] | 19 (1.70) | 2 (0.4) | 18 (1.71) |
| APOE ε4 alleles [n (%)] | 0 | 542 (47.5) | 256 (49.3) | 516 (49.0) |
| | 1 | 517 (45.4) | 214 (42.9) | 461 (43.8) |
| | 2 | 81 (7.1) | 29 (5.81) | 76 (7.21) |
| MMSE [mean (sd)] | | 28.3 (1.89) | 28.3 (1.81) | 28.3 (1.91) |
| ADAS-cog [mean (sd)] | | 3.74 (2.73) | 3.76 (2.78) | 3.69 (2.74) |
| Aβ PET, SUVR [mean (sd)] | | 1.10 (0.303) | 1.09 (0.30) | 1.10 (0.299) |
| Aβ PET, status [n (%)] | Normal | 717 (62.9) | 337 (67.5) | 674 (64.0) |
| | Abnormal | 423 (37.1) | 162 (32.5) | 379 (36.0) |
| Plasma %P-tau217 [mean (sd)] | | | 1.34 (1.30) | 1.32 (1.21) |
| Plasma Aβ42/Aβ40 [mean (sd)] | | 0.118 (0.0110) | 0.118 (0.0110) | 0.118 (0.111) |
| CSF P-tau217, pg/mL [mean (sd)] | | | 13.8 (19.4) | |
| CSF Aβ42/Aβ40 [mean (sd)] | | 0.0903 (0.0319) | 0.0920 (0.0321) | 0.0907 (0.0318) |

The table shows the overall study population ("BF2"). Initial model development was done on a subset with no missing data, "BF2-Initial". For further analyses, distinct sub-cohorts were defined and used, differentiated by the availability of P-tau217 measurements in plasma and CSF. The rest of the main analyses were done on individuals with available plasma %P-tau217 (mass spectrometry data; "BF2-P-MS"). Other subgroups used in the paper are described in Table EV1. Note that P-tau217 is quantified in varying units across these sources, with WashU using percent mean and Lilly using picograms per milliliter (pg/ml).
[a]Dementia included AD (n = 10), Frontotemporal (n = 3), Parkinson (n = 1), Progressive supranuclear palsy (n = 2), Vascular (n = 1), and not determined underlying etiology (n = 2).

## Initial feature selection

Followed by the initial model selection, we trained an Extra Trees Regressor and analyzed the feature importance using SHAP. As depicted in Fig. 1, we listed the top 15 features with the highest SHAP impacts, in which CSF and plasma biomarkers were the most influential, especially CSF Aβ42/Aβ40 and plasma P-tau217 assays (out of which %P-tau217$_{WU}$ had highest SHAP impact). The rest of the biomarkers, demographic factors and cognitive tests had less influence. The feature importance was overall similar across three different methods, including impurity-based importance (the embedded method within the Extra Trees Regressor), permutation importance and SHAP analysis (Appendix Figs. S1–S3).

## Final feature selection

The final selection of features was based on a combination of information from the SHAP values, and control experiments of information redundancy from different features (we evaluated models with and without selected features, Table EV2). We noted that in combination with CSF Aβ42/Aβ40 and age, the three plasma P-tau217 assays plasma %P-tau217, P-tau217$_{WU}$ and P-tau217$_{Li}$ performed similarly, with a small advantage for %P-tau217 (see feature combinations 1, 2 and 3). CSF P-tau217 also reached similar

performance, but slightly less than all the three plasma P-tau217 biomarkers (feature combination 4). The influence of plasma Aβ42/Aβ40 was considerably lower than CSF Aβ42/Aβ40 (compare feature combinations 6 and 8). There were no signs of plasma Aβ42/Aβ40 improving upon a model with plasma %P-tau217 (compare feature combinations 5 and 7). In contrast, CSF Aβ42/Aβ40 contributed to the predictions also in the presence of plasma %P-tau217 (compare feature combinations 1 and 5). Whenever plasma %P-tau217 was included, adding also P-tau231, %P-tau205, and %P-tau181 did not improve or only very marginally improved performance (see feature combinations 1 and 5 compared to 10 and 9, respectively). Adding cognitive and demographic factors (e.g., cognitive status and cognitive tests) resulted in a minor further increase beyond using a comprehensive set of biomarkers (compare feature combinations 10 and 11).

One objective of this study was to evaluate the predictive capabilities of models using cutting-edge plasma biomarkers for AD to predict continuous Aβ-PET SUVR values, considering a trade-off between model complexity and predictive performance. As outlined above, we identified two critical biomarker features to be the most informative: (plasma or CSF) P-tau217 and (CSF) Aβ42/40. In our primary analysis, we therefore used plasma %P-tau217 measured with MS, together with CSF Aβ42/Aβ40 (taken to represent the best possible predictions from a fluid measure of

**Table 2. Demographics of participants from the Swedish BioFINDER-1 Study.**

| | | BF1-P-IA |
|---|---|---|
| *n* | | 238 |
| Age [mean (sd)] | | 72.3 (5.5) |
| Sex | Male | 118 (49.6) |
| | Female | 120 (50.4) |
| Education [mean (sd)] | | 11.6 (3.3) |
| Cognitive status [*n* (%)] | CN | 73 (30.7) |
| | SCD | 70 (29.4) |
| | MCI | 95 (39.9) |
| | Dementia | 0 (0.0) |
| *APOE* ε4 alleles [*n* (%)] | 0 | 133 (55.9) |
| | 1 | 85 (35.7) |
| | 2 | 20 (8.4) |
| MMSE [mean (sd)] | | 28.1 (1.6) |
| ADAS-cog [mean (sd)] | | 4.4 (2.8) |
| Aβ PET, SUVR [mean (sd)] | | 1.2 (0.3) |
| Aβ PET, status [*n* (%)] | Normal | 125 (52.5) |
| | Abnormal | 113 (47.5) |
| Plasma P-tau217, pg/ml [mean (sd)] | | 0.2 (0.2) |
| CSF Aβ42/Aβ40 [mean (sd)] | | 0.07 (0.03) |

All the BF1 participants used in this analysis had plasma P-tau217 measurements by Lilly immunoassay (BF1-P-IA).

Aβ42/Aβ40). We also evaluated a plasma-only workflow using only plasma %P-tau217. In addition to the fluid biomarkers, we included age given its high availability and its relatively high impact on the predictions according to the SHAP analysis.

## Performance of the optimal predictive model—plasma mass spectrometry %P-tau217, CSF Aβ42/Aβ40 and age

As outlined above, the chosen model architecture for producing all main results was the Extra Trees Regressor, with three key features: plasma %P-tau217, CSF Aβ42/Aβ40 and age. We trained an Extra Trees Regressor using 80% of the sub-cohort with plasma %P-tau217 (BF2-P-MS) data from BioFINDER-2, and tested on a 20% test subset. Observed Aβ-PET data versus predicted Aβ-PET data, and model residuals, are shown in Fig. 2A. The errors increased with higher Aβ-PET load. The model predictions had low variability within the negative range of Aβ-PET (Fig. 2B). The overall $R^2$ was 0.79 and the overall MAPE (Mean Absolute Percentage Error) was 7%, on the whole Aβ-PET range in the test set (Fig. 2C). We also separately evaluated the errors within the negative and positive range of Aβ-PET, which amounted to 5% in the negative range and 11% in the positive range. Furthermore, we assessed the practical applicability of the regression model, by computing the probabilities of a predicted value representing various underlying observed values (Fig. 2D; Appendix Fig. S4).

## Investigating feature contributions of plasma %P-tau217 and CSF Aβ42/40

While evaluating the feature importance of the regression models, using different machine learning models, we noted fluctuations of

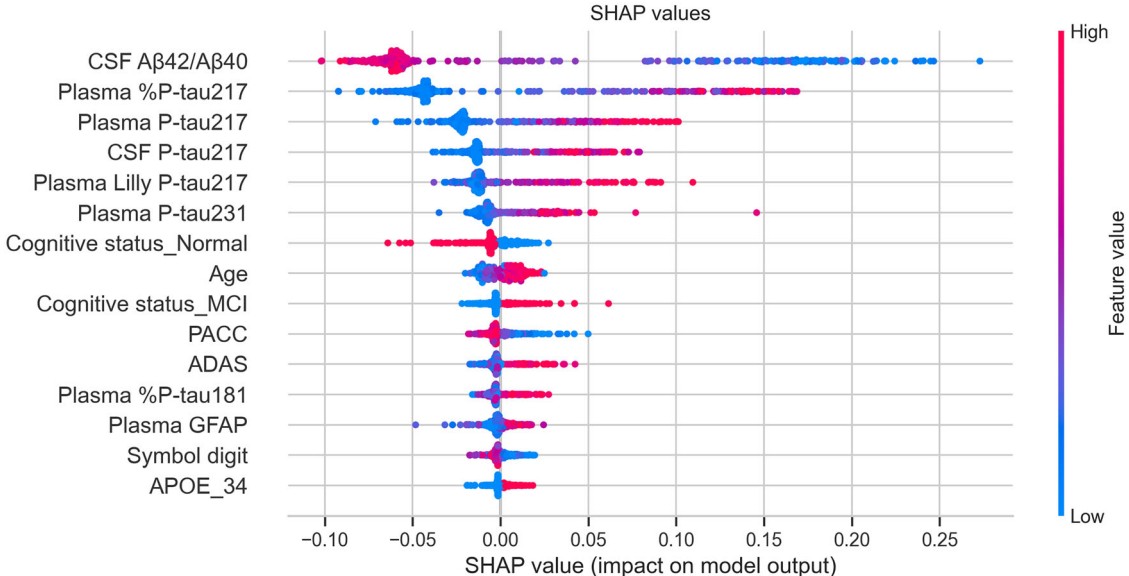

**Figure 1. Analysis of feature importance.**

The prediction of Aβ-PET was inferenced by the Extra Trees Regressor trained on BF2-Initial. The Y-axis represents the input features, while the X-axis displays the corresponding SHAP values for the top 15 highest impacts (note that plasma Aβ42/Aβ40 was not in the top 15 and is therefore not shown). Features with positive SHAP values and lower feature values are inversely related to the model's predictions, and vice versa. The magnitude of the SHAP value directly correlates with the feature's significance, where greater absolute values signify higher influence on the model's output. Source data are available online for this figure.

 

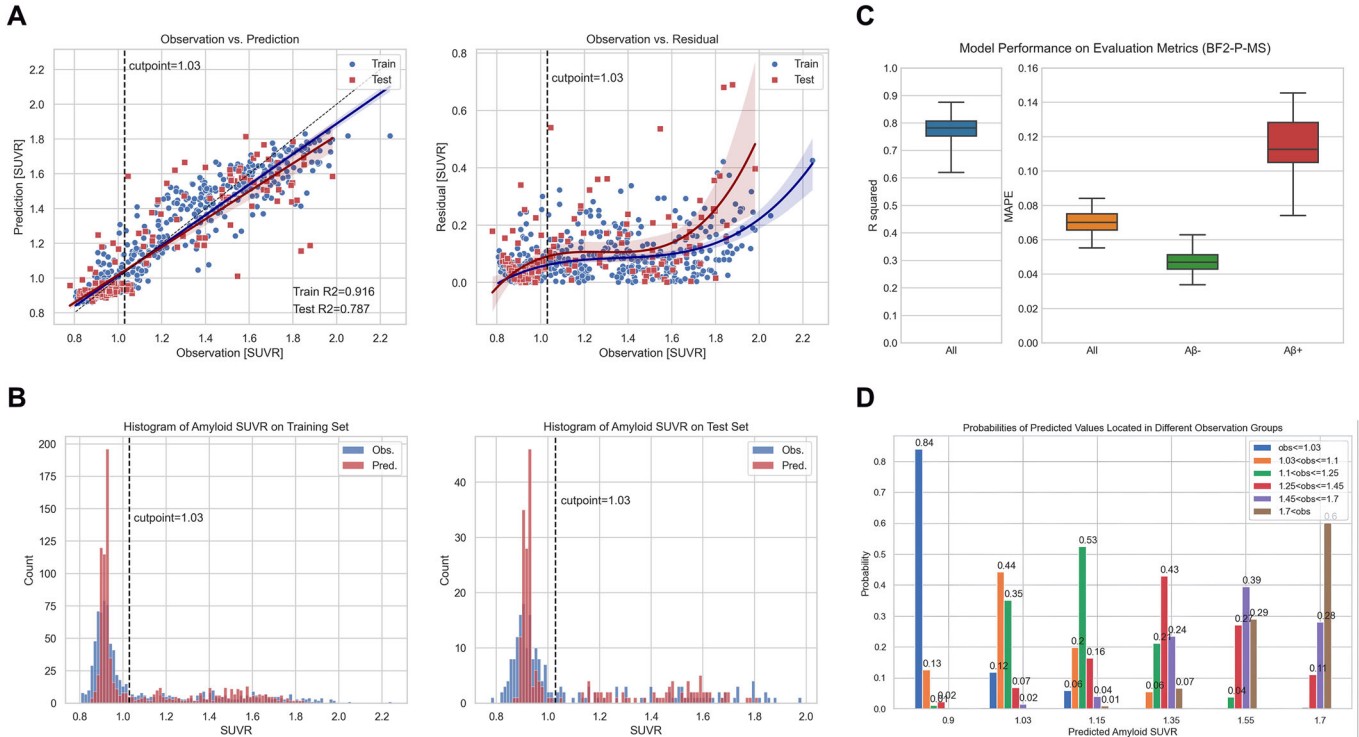

**Figure 2. Performance of the best predictive model (Extra Trees Regressor) trained and tested on BF2-P-MS for the %P-tau217, CSF Aβ42/Aβ40 and age model.**

(**A**) Observations versus predictions and residuals. (**B**) Distributions of observed and predicted Aβ-PET SUVR on training and test set, respectively. (**C**) Results on the test set with $R^2$ and MAPE on all individuals, and stratified by Aβ-PET negativity and positivity. The results were generated using a bootstrapping method where we resampled 80 samples with replacement for 100 iterations from the test set in order to evaluate the variance of the test set. The box plots show the quartiles of these 100 iterations (IQR = Q3–Q1), with median as center line, and whiskers extending to minimum and maximum points. (**D**) Probabilities of the model's predictions (6 specific cut points) falling within a number of observation groups, with further details in Appendix Fig. S9. Source data are available online for this figure.

the results (Appendix Fig. S4). The long-tail distribution of observed Aβ-PET values shows that most samples are concentrated in the Aβ-negative range. The low variance of predicted Aβ-PET values in this range may introduce challenges in achieving consistent and interpretable results for feature importance. To address this issue while ensuring that the data range remained unchanged (including both Aβ-negative and Aβ-positive values), we proceeded to construct a classifier to divide Aβ values into negative and positive groups. This allowed us to evaluate the feature importance and its stability when the complexity of the training problem was reduced. We next segregated Aβ-negative and Aβ-positive samples into separate groups, and then each Aβ group was fed into an untrained regression model for training separately. For simplicity, we referenced these two regressors as the Aβ-negative and Aβ-positive regression models, respectively, in the following sections. These procedures facilitated easier and clearer evaluation of feature importance across different Aβ statuses. To create a more suitable environment for comparing the two influential biomarkers, we excluded age from this analysis.

The Aβ-negative regression model achieved an $R^2$-value of 0.06 on the test set (Fig. 3A). For the Aβ-positive group, the regression model attained an $R^2$-value of 0.56 on the test set (Fig. 3B). The classifier obtained an AUC of 0.98 on the test set (Fig. 3C). Given the inferior $R^2$ obtained by the Aβ-negative regression model, we did not consider it meaningful to further investigate the feature importance in the

negative range. Figure 3D presents the weighted feature importance (scaled so in total 100%) of the Aβ-positive regression model and the classifier. Within the Aβ-positive range, plasma %P-tau217 was the dominant predictor with a feature importance of approximately 75%. In the task of classification, CSF Aβ42/40 was the most influential biomarker with a feature importance of approximately 60%.

We also conducted an analysis where age and *APOE* genotype were also included, and the only significant change was that 'age' contributed nearly 60% to the prediction in the Aβ-negative regression test, with the $R^2$-score marginally increased to 0.1 (Appendix Fig. S5). This aligns with the notion that Aβ load is challenging to measure without noise in Aβ-negative participants, with 'age' primarily reflecting the lifetime accumulation and increased likelihood of amyloid plaques in older participants. Another possibility is that the relationship between fluid biomarkers and PET changes with age.

## Performance of the optimal predictive model—plasma mass spectrometry %P-tau217 and age only

We next investigated a plasma-only workflow, which would be a clinically relevant approach reducing cost and invasiveness of CSF sampling. We trained an Extra Trees Regressor using plasma %P-tau217 and age using 80% of the sub-cohort (BF2-P-MS) and tested it on the left-out 20% subset. Observed Aβ-PET data versus

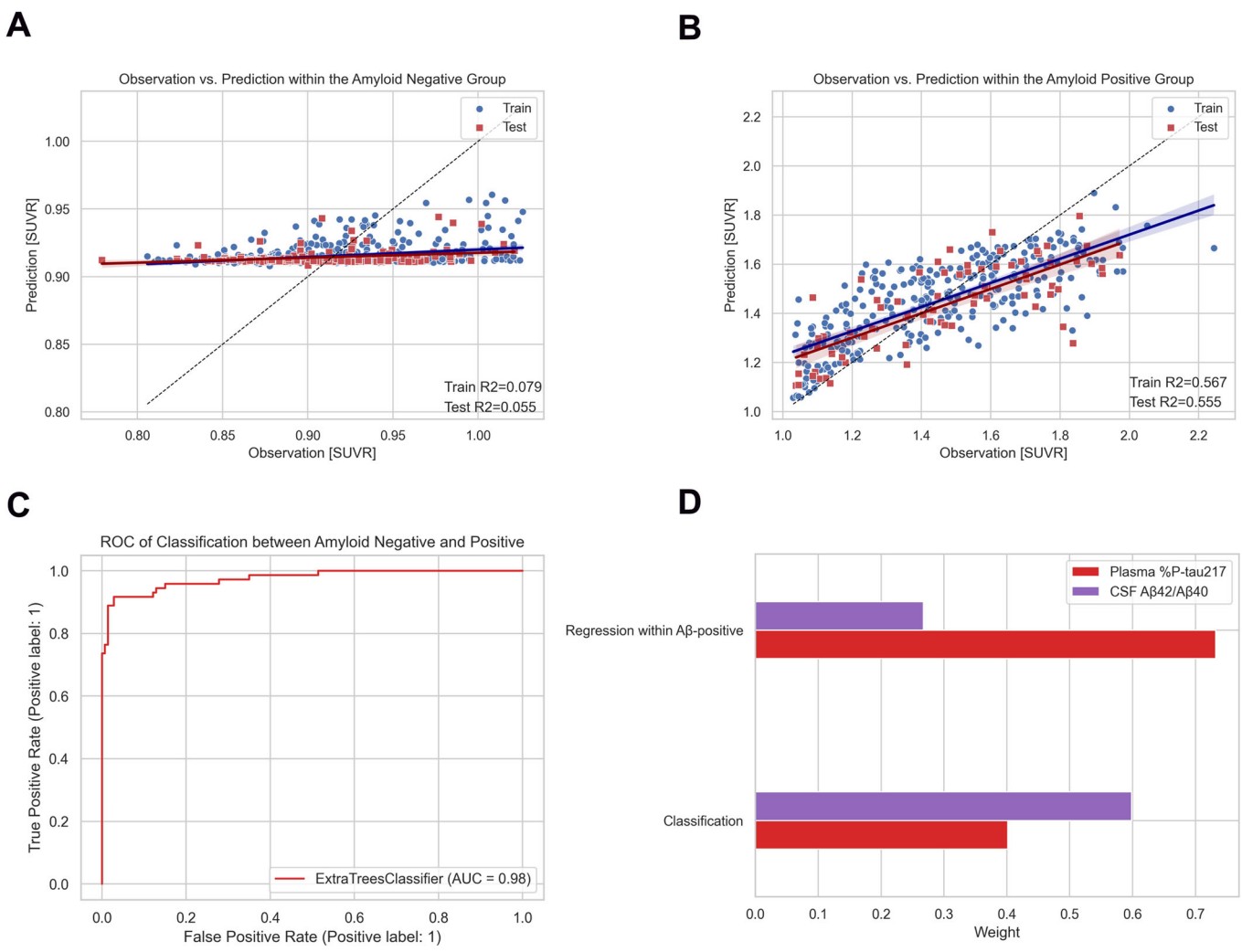

**Figure 3. A two-step procedure for clearer analyses of feature importance.**

(A, B) Observation versus prediction within the Aβ-negative and positive group, respectively (note that separate models were trained for the two groups). (C) Receiver operating characteristic curve for the binary classifier in the test set. (D) The feature importance for both the classifier and the positive regressor. Source data are available online for this figure.

predicted Aβ-PET data, and model residuals, are shown in Fig. 4A. Similar to the %P-tau217, CSF Aβ42/Aβ40 and age model, the errors increased with higher Aβ-PET load. The overall $R^2$ was 0.73 and the overall MAPE was 8%, on the whole Aβ-PET range in the test set (Fig. 4B, compared to $R^2 = 0.79$ and MAPE = 7% for the model also including CSF). Separately evaluating the errors within the negative and positive Aβ-PET ranges in the test set resulted in 6% in the negative range and 12% in the positive range (compared to 5% and 11% for the model also including CSF).

## Model generalization assessment in BioFINDER-1

To assess the prediction generalizability, we evaluated the best model in the external cohort BioFINDER-1 (BF1). BF1 did not include measures of plasma %P-tau217 with MS, but only P-tau217 IA (Lilly), so we retrained the Extra Trees Regressor model using BF2-P-IA with plasma P-tau217 IA (Lilly), CSF Aβ42/Aβ40 and

age. The results revealed that the model exhibited comparable but on average slightly lower performance in BF1 compared to BF2 test (Fig. 5A,B; $R^2 = 0.67$ and MAPE = 0.1 in BF1 and $R^2 = 0.77$ and MAPE = 0.07 in BF2 test).

Moreover, in the two-step strategy, the regression results within the Aβ-positive range were further validated in BF1 with similar results (Appendix Fig. S6).

## Sensitivity analyses

We performed a sensitivity analysis for the main results when using another machine learning model, the GradientBoost regressor, to show that the results are not dependent on the structure (e.g., Extra Trees Regressor) of the model itself. The relevant results are displayed in Appendix Figs. S7 and S8.

We also applied the entire pipeline to the other sub-cohorts sourced from BioFINDER-2, utilizing plasma P-tau217 data

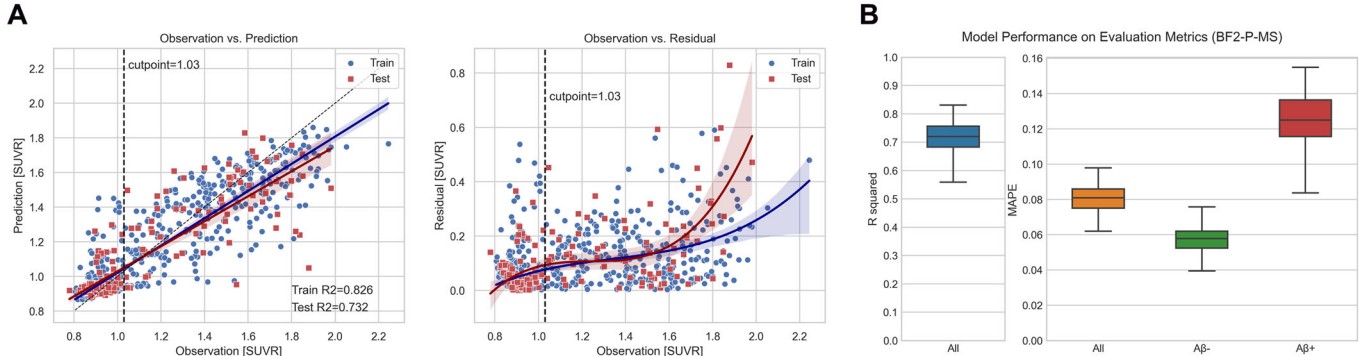

**Figure 4.  Performance of the best predictive model (Extra Trees Regressor) trained and tested on BF2-P-MS for the %P-tau217 and age model.**

(A) Observations versus predictions and residuals. (B) Results on the test set with $R^2$ and MAPE on all individuals and stratified by Aβ-PET negativity and positivity. The results were generated using a bootstrapping method where we resampled 80 samples with replacement for 100 iterations from the test set to evaluate the variance of the test set. The box plots show the quartiles of these 100 iterations (IQR = Q3–Q1), with median as center line, and whiskers extending to minimum and maximum points. Source data are available online for this figure.

obtained by Lilly immunoassay (BF2-P-IA; Fig. EV1A), and CSF P-tau217 data analyzed by either MS-method (BF2-C-MS; Fig. EV1C) or the Lilly immunoassay (BF2-C-IA; Fig. EV1B). A comparison between plasma P-tau217 and CSF P-tau217 revealed that models trained with CSF P-tau217 exhibited relatively poorer overall performance (but note that models tuned for plasma may not be expected to translate perfectly to CSF). Specifically, the mean $R^2$ scores averaged around 0.74 and 0.72 for BF2-C-IA and BF2-C-MS, respectively, in contrast to the mean $R^2$ scores of 0.83 and 0.78 for BF2-P-MS and BF2-P-IA.

## Discussion

This study investigated the association between fluid biomarker levels and continuous Aβ-PET data, in a population with the continuum from cognitively normal to patients with cognitive impairment. Our main goal was to define an optimal model to predict Aβ-PET with fluid biomarkers and identify the most important biomarker features for predictions in different ranges of Aβ-PET, to gain in vivo biological insights about links between fluid biomarkers and Aβ aggregation. Our pipeline combined a data driven approach with a priori information about key measures relevant to prediction of Aβ-PET. We developed a two-stage approach, where Aβ-PET negative individuals were first ruled out by a classifier, and then the Aβ-PET load was estimated within the positive range by a regressor. Classification of Aβ-negative and Aβ-positive could be performed with high accuracy with AD-related fluid biomarkers, as has also been shown before (Palmqvist et al, 2014; Janelidze et al, 2016; Mattsson-Carlgren et al, 2023) (and which is higher than reported for models using more basic data from routine clinical visits (Kimura et al, 2025)). In contrast, association between fluid biomarkers and continuous Aβ-PET within the positive range, was a more difficult task. Almost 50% of the variability in Aβ-PET in the positive range remained unaccounted for even in our best fluid biomarker models. Two biomarkers stood out as the most informative, with complimentary information: CSF Aβ42/Aβ40 (most important to classify Aβ-negatives versus positives) and plasma P-tau217 (most important to

estimate Aβ-PET within the positive range). Taken together, our results demonstrate that specific fluid biomarkers can partly substitute for Aβ-PET and emphasize that altered metabolism of soluble Aβ is a key feature of the overall presence of Aβ pathology, while altered metabolism of soluble P-tau is related to the magnitude of Aβ pathology within the AD continuum.

Our best model achieved an overall $R^2$ of 0.79 to predict Aβ-PET SUVR. This was for a model that included both CSF Aβ42/Aβ40 and plasma %P-tau217, but when excluding CSF, the $R^2$ only dropped to 0.73 (out-of-sample evaluation). The finding that a fluid biomarker-based model (even when restricted to plasma %P-tau217) can explain >70% of the variance in overall Aβ-PET SUVR is encouraging for the use of fluid biomarkers as a substitute to Aβ-PET for some applications, for example for enrichment in clinical trials, or for a first stage triaging of subjects that can be funneled to more advanced testing. The findings are in line with results by Devanarayan et al, who also reported plasma %P-tau217 showing highest influence during Aβ-PET prediction compared to plasma Aβ42/Aβ40, with a similar but slightly lower $R^2$ of 0.63–0.66 compared to our %P-tau217 model ($R^2 = 0.73$) (Devanarayan et al, 2024).

When examining model performance within the Aβ-negative and Aβ-positive ranges separately, we found that values in the Aβ-negative range were almost unpredictable even with the best fluid biomarkers ($R^2 < 0.1$). The variability in Aβ SUVR in the negative range largely represents noise, which is not reflected by independent fluid biomarkers. We therefore developed a two-step strategy with an initial classification of Aβ-negative versus Aβ-positive, followed by regression in the positive range. The classifier had high performance to differentiate Aβ-negative versus Aβ-positive, with AUC of 0.98. This is similar to what has been reported for other fluid biomarkers to classify Aβ-PET-status, across a range of thresholds for Aβ-PET positivity (Milà-Alomà et al, 2022; Barthélemy et al, 2024; Mattsson-Carlgren et al, 2023; Rissman et al, 2023). A regressor within the Aβ-negative range had out-of-sample $R^2 = 0.06$, confirming that precise prediction of Aβ PET-SUVR in the negative range is not feasible. In contrast, the model had $R^2 = 0.56$ in the Aβ-positive range, which we consider to be a moderate correlation. The relatively lower $R^2$ scores for Aβ PET in

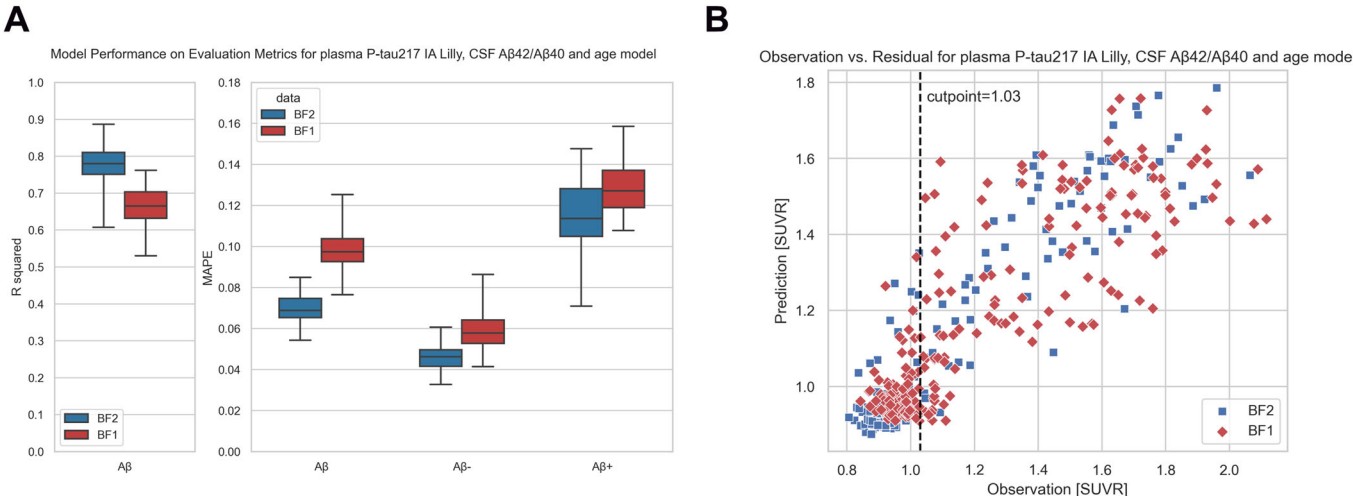

**Figure 5. Validation results in the external cohort BioFINDER-1.**

We retrained an Extra Trees Regressor on the BF2 training set with P-tau217 IA Lilly, CSF Aβ42/Aβ40 and age and compared the prediction results on the test set from BF2 and the external cohort BF1. In (A), corresponding summary metrics for BF2 test and BF1 are shown. The results were generated using a bootstrapping method where we resampled 80 samples with replacement for 100 iterations from the test set/BF1 in order to evaluate the variance of the test set/BF1. The box plots show the quartiles of these 100 iterations (IQR = Q3–Q1), with median as center line, and whiskers extending to minimum and maximum points. In (B), observations versus predictions for both BF2 test and BF1. Source data are available online for this figure.

the negative and positive ranges do not contradict the high R² score of 0.79 across the entire Aβ-PET SUVR range. This is because the regressor trained on the entire range captures significant covariance between Aβ-negative and Aβ-positive samples, which is a crucial component of the total variance. In other words, while the original model struggles to achieve stable and accurate predictions in the Aβ-negative range, it has effectively learned some aspects of the classification achieved by the classifier in the two-step strategy. For better interpretability of the model performance, we computed the probability that a given predicted Aβ PET-SUVR corresponded to a true underlying Aβ PET-SUVR (Fig. 3D). For example, for a predicted Aβ-PET SUVR of 1.35, the probability was ~45% that the underlying true Aβ-PET was in the range 1.25–1.45 SUVR, with lower probabilities for tails of true Aβ-PET values (~21% for 1.11–1.25 SUVR and ~24% for 1.45–1.70 SUVR). These transparent results are somewhat humbling for our capacity to accurately predict Aβ-PET burden with fluid biomarkers, and shows that there remains a great need for novel fluid biomarkers that reflect the actual Aβ-burden with greater precision.

Among all tested biomarkers, the strongest contributions came from CSF Aβ42/Aβ40 and plasma %P-tau217 (closely followed by total plasma P-tau217 from mass spectrometry, in line with another publication showing similar performance for %P-Tau217 and P-tau217 to predict Aβ positivity (Barthélemy et al, 2024)). Other biomarkers provided only marginal improvement or redundant information when included in addition to these two biomarkers. Aβ42/Aβ40 and %P-tau217 provided differential information for different aspects of Aβ-PET. While CSF Aβ42/Aβ40 was the most informative measure to differentiate Aβ-negative from Aβ-positive participants, plasma %P-tau217 provided most information about the continuous Aβ-PET load within the positive range. The finding that CSF Aβ42/Aβ40 was informative primarily for classification is well in line with previous reports showing high overall classification

concordance for these measures (Janelidze et al, 2016; Keshavan et al, 2021), with only minor variability in CSF Aβ42/Aβ40 among individuals with positive Aβ-PET. CSF Aβ42 is reduced and reaches a plateau while Aβ-PET continues to increase (Mattsson et al, 2019b). CSF Aβ40 is typically not altered in AD, but serves as a reference peptide which sharpens the performance of CSF Aβ42 (Karlsson et al, 2024). Our results support the notion that the presence of Aβ pathology is associated with a change in the metabolism of soluble Aβ42, which reaches an equilibrium early in the disease process, and do not directly reflect the amount of Aβ pathology in the brain, which also is in agreement with the know distinct bimodal distribution of CSF Aβ42/Aβ40 values even in unselected patient cohorts (Gobom et al, 2022). Plasma Aβ42/Aβ40 had less influence in our models (especially when plasma %P-tau217 was included), which is in agreement with a previously described inferior performance of Aβ42/Aβ40 in plasma compared to in CSF (likely caused by peripherally produced Aβ peptides produced in plasma giving an interference resulting in smaller fold-change for Aβ42/Aβ40 in plasma compared to in CSF for positive Aβ-status) (Schindler et al, 2019). Plasma %P-tau217 contributed less to the classifier of Aβ-status (when CSF Aβ42/Aβ40 was in the model) but dominated the prediction of continuous Aβ-uptake in the positive range. Previous studies have shown positive correlations between P-tau217 (in plasma or CSF) and Aβ-PET in the positive range (Milà-Alomà et al, 2022; Mattsson-Carlgren et al, 2021, 2020), but have also shown that P-tau217 has excellent properties as a biomarker to identify Aβ-positivity (Barthélemy et al, 2024). With the simultaneous assessment of Aβ42/Aβ40 and P-tau217 for both classification of Aβ-status and regression within the positive range of Aβ, we can now clarify the predominant and independent capacities of these biomarkers. The relationship between P-tau217 and Aβ-pathology is complex. We have previously shown that P-tau217 is associated with continuous

Aβ-PET burden also among those with positive Aβ-PET but negative tau PET, indicating a quite early stage of AD (Mattsson-Carlgren et al, 2021). On the other hand, in individuals who were positive for both Aβ-PET and tau PET, P-tau217 was predominantly associated with tau PET. It is possible that P-tau217 reflects both the Aβ-burden and tau burden in AD. This is also supported by CSF studies showing a distinct association between P-tau217 (and P-tau181) and evolution of tau PET positivity with early increases preceding tau PET positivity (Mattsson-Carlgren et al, 2020), and neuropathology studies, which have linked increased plasma P-tau217 to both amyloid plaques and tau tangles (while Aβ42/Aβ40 was much more strongly associated with amyloid plaques) (Salvadó et al, 2023). Potentially, the presence of Aβ pathology (which is a necessary feature for the subsequent development of aggregated tau pathology) induces changes in the metabolism of phosphorylated tau, leading to increased levels of P-tau217 in a manner proportional to the degree of Aβ pathology (while other tau-related fluid biomarkers are more closely linked to the aggregated tau burden (Horie et al, 2023)).

One limitation of the study was the varying availability of biomarkers, which reduced the sample size for experiments that involved several biomarkers at once. However, results were generally consistent across analyses in subgroups (e.g., for biomarkers in CSF, or plasma, or biomarkers assessed with different analytical methods). A related limitation was that due to the limited data, model performance was highly sensitive to the test set distribution. To mitigate the risk of an unrepresentative test set during data splitting, we evaluated the model's performance using various random seeds and combined the results with the bootstrapping resampling method. This allowed us to robustly assess the model's performance and the internal variance of the test set. Third, while we focused on a range of well-established models and rigorously optimized their features and hyperparameters, future work could explore more advanced algorithms, such as deep learning architectures (which may need larger sample sizes for efficient training), to potentially further improve predictive performance. A fourth limitation was that, according to the BF2 study protocol, Aβ-PET was rarely performed in individuals with dementia, which limited our ability to evaluate model performance at this disease stage. Certain features, such as cognitive status, may have contributed more strongly to prediction if a larger number of dementia cases had been included. In addition, inclusion in BF2 was stratified by APOE ε4 status among cognitively unimpaired individuals, which may have influenced the contribution of this feature compared to a population with frequencies closer to global prevalence. Another limitation was the relatively homogenous genetic and environmental background of BF2, factors that could potentially influence biomarker performance. Even though the models generalized well to the independent cohort BF1, findings should be validated across more diverse populations in future studies. A final limitation was the cross-sectional design. Longitudinal studies with parallel repeated measures of both fluid biomarkers and Aβ-PET are needed to clarify if fluid biomarker-based models can also be used to track the Aβ-burden over time with similar precision as in the cross-sectional setting. This will be especially important in the context of treatments targeting Aβ, where studies should clarify that any changes in fluid biomarkers are proportional to changes in Aβ PET uptake.

We conclude that fluid biomarkers may predict Aβ PET-uptake, with CSF Aβ42/Aβ40 providing most information about Aβ PET-status (positive or negative), while plasma or CSF P-tau217 provides most information about the load of Aβ PET-burden within the positive range. These findings may be used to empower models of Aβ PET-burden with fluid measures. The findings may also inform about the use of these fluid biomarkers as outcome measures in interventional trials, where Aβ42/Aβ40 and p-tau217 may reflect different aspects of Aβ pathology.

# Methods

**Reagents and tools table**

| Reagent/Resource | Reference or Source | Identifier or Catalog Number |
|---|---|---|
| **Antibodies** | | |
| p-tau217 immunoassay | Lilly Research Laboratories | |
| Elecsys β-amyloid(1–42), β-amyloid(1–40) and p-tau181 immunoassays | Roche Diagnostics | |
| Simoa P-tau231, NTA, GFAP, and NFL Discovery Kits | Quanterix/University of Gothenburg | |
| **Chemicals, Enzymes and other reagents** | | |
| Liquid chromatography–tandem mass spectrometry (LC–MS/MS) | Department of Neurology, Washington University School of Medicine | |
| [18 F]Flutemetamol | GE Healthcare | |
| **Software** | | |
| Python | Python Software Foundation | v.3.9 |
| **Other** | | |
| BioFINDER-2 cohort | Lund University/Skåne University Hospital | NCT03174938 |
| BioFINDER-1 cohort | Lund University/Skåne University Hospital | NCT01208675 |
| GE Discovery MI PET/CT scanner | GE Healthcare | |

## Participants

The study participants came from two independent prospective cohort studies: Swedish BioFINDER-1 (NCT01208675) and BioFINDER-2 (NCT03174938). Participants in these cohorts were recruited at Skåne University Hospital and Ängelholm Hospital in Sweden, and included cognitively unimpaired (CU) individuals (combining cognitively normal older adults and individuals with subjective cognitive decline (Jack et al, 2018)), and patients with mild cognitive impairment (MCI) or dementia. Detailed inclusion and exclusion criteria have been described before (Palmqvist et al, 2020, 2023, 2021; Quadalti et al, 2023). Briefly, the participants were aged 40–100 years, and spoke and understood Swedish to the extent that an interpreter was not necessary. All participants were enrolled and underwent baseline examination either from 2011 to

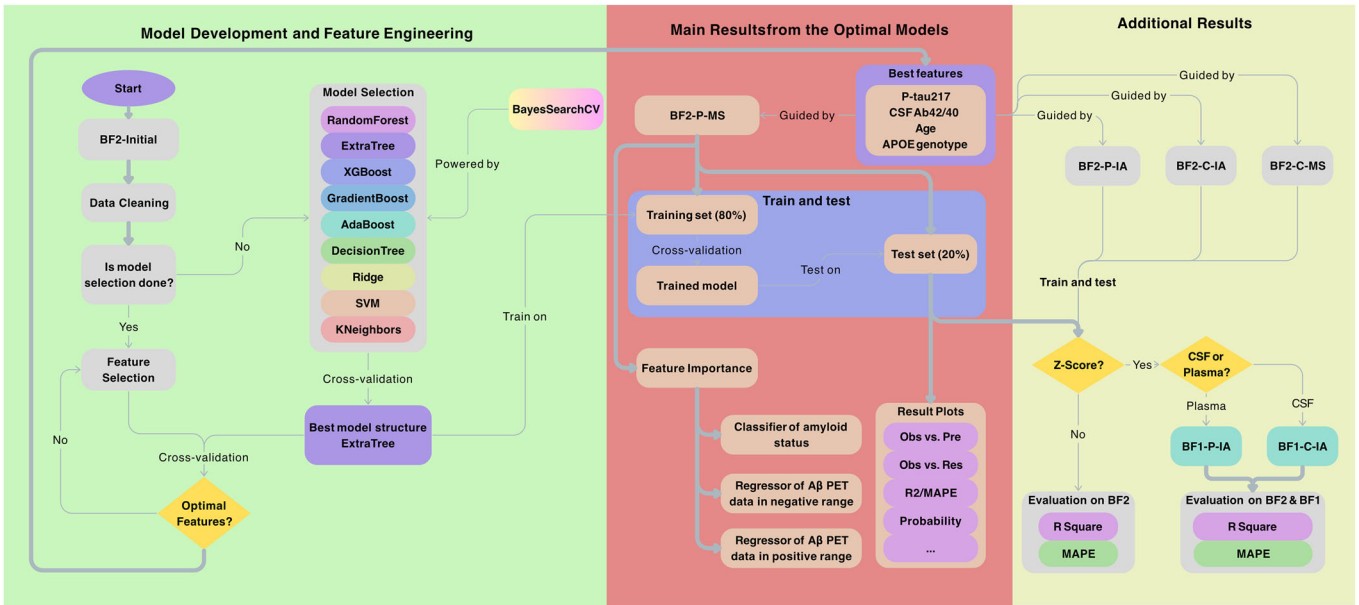

**Figure 6. The overall workflow of the analysis.**

The left part of the flow chart (highlighted in green) shows the Model Development and Feature Engineering, where the BF2-P-MS sub-cohort was used as the primary dataset. Model selection for the regression problem was done across a range of potential regressors, identifying Extra Trees as the best model structure. Feature selection by SHAP was done with cross-validation, to obtain an optimal model. The middle part of the flow chart (highlighted in red) shows the Main Results from the Optimal Models. The results come both from regression analysis on the whole range of Aβ-PET SUVR and from independent assessment of feature importance when using a classifier (for positive versus negative Aβ-PET) followed by two additional regressors (within the negative and positive range of Aβ-PET). The right part of the flow chart (highlighted in yellow) shows Additional Results for training and testing the P-tau217 with different sources in sub-cohorts from BioFINDER-2 (BF2-P-IA, BF2-C-IA, BF2-C-MS), and evaluation of the model's generalizability in BioFINDER-1 (BF1-P-IA and BF1-C-IA). BF1 BioFINDER-1, BF2 BioFINDER-2, C-IA cerebrospinal fluid P-tau217 data by immunoassay, C-MS cerebrospinal fluid P-tau217 data by mass spectrometry, MAPE mean absolute percentage error, P-IA plasma P-tau217 data by immunoassay, P-MS plasma P-tau217 data by mass spectrometry.

2018 (BioFINDER-1) or from 2017 to 2024 (BioFINDER-2). All participants gave written informed consent. Ethical approval was given by the Swedish Ethical Review Authority. Approval for PET imaging was obtained from the Swedish Medicines and Products Agency and the local Radiation Safety Committee at Skåne University Hospital in Sweden.

## Cerebrospinal fluid (CSF) and plasma biomarkers

Plasma and CSF levels of P-tau181, and P-tau217, together with plasma levels of P-tau205 were measured in BioFINDER-2 using liquid chromatography-tandem mass spectrometry (LC-MS/MS) at the Department of Neurology, Washington University School of Medicine (Barthélemy et al, 2020). Levels are reported here both as P-tau181, P-tau205 and P-tau217 (for absolute levels), and as %P-tau181, %P-tau205, and %P-tau217 (for levels relative to corresponding non-phosphorylated tau isoforms, multiplied by 100). Plasma (as well as CSF) levels of P-tau217 were also determined in both BioFINDER-1 and BioFINDER-2 at Lund University using an immuno-assay developed by Lilly Research Laboratories (Mattsson-Carlgren et al, 2021). Note that the calibration methods for the P-tau217 assay varied by cohort (BioFINDER-1 used a synthetic peptide for P-tau217, whereas BioFINDER-2 employed an in vitro phosphorylated recombinant tau protein), resulting in different value ranges. To facilitate comparisons between the cohorts, levels of P-tau217 were therefore standardized to z-scores (using the mean and standard deviation in

Aβ-negative CU in each cohort). CSF levels of Aβ40, Aβ42 and P-tau181 were assessed using Roche Elecsys immuno-assays (Mattsson-Carlgren et al, 2022), in both cohorts. In the BioFINDER-2 cohort, plasma levels of P-tau231, NTA (N-terminal containing tau fragments), GFAP and NFL were measured using in-house Simoa immuno-assays (Ashton et al, 2021; Lantero-Rodriguez et al, 2024) developed at the University of Gothenburg and commercially available Simoa Discovery immuno-assay (Quanterix) (Pereira et al, 2021; Mattsson et al, 2019a), while plasma Aβ42 and Aβ40 levels were quantified using LC-MS/MS at Washington University (Schindler et al, 2019; Janelidze et al, 2021). CSF and plasma analyses were performed by technicians blinded to all clinical and imaging data.

Initial model development (comparison of different regressor variants) was done on a subset of BioFINDER-2 with no missing data ("BF2-Initial"). More detailed model development in our machine learning pipeline was done on BioFINDER-2 subjects with plasma P-tau217 measured with LC-MS/MS (this subset is called "BF2-P-MS" below). Other main subsets used in different parts of the analysis are "BF2-P-IA" (BioFINDER-2 subjects with plasma P-tau217 measured with immunoassay), "BF2-C-IA" (BioFINDER-2 subjects with CSF P-tau217 measured with immunoassay) and "BF2-C-MS" (BioFINDER-2 subjects with CSF P-tau217 measured with LC-MS/MS). Independent BioFINDER-1 subjects are used only for validation purpose, including "BF1-P-IA" (BioFINDER-1 subjects with plasma P-tau217 measured with immunoassay), "BF1-C-IA" (BioFINDER-1 subjects with CSF P-tau217 measured

with immunoassay). Details of these sub-cohorts are presented in Tables 1, 2 and EV1.

## Aβ-PET acquisition and processing

Aβ-PET imaging was performed on digital GE Discovery MI scanners, with 4 frames of 5 min acquired 90–110 min post-injection of ~185 MBq [$^{18}$F]Flutemetamol. Global SUVR values were calculated for a composite region, with positive Aβ-PET status determined at a pre-specified cut-off of 1.03 SUVR (Ossenkoppele et al, 2022).

## Cognitive assessments

We used the Mini-mental state examination (MMSE) as a measure of global cognition. The ten-word delayed recall test from the Alzheimer's Disease Assessment Scale-Cognitive Subscale (ADAS-Cog) was used as a measure of memory. The Trail-Making Test A (TMT-A), Trail-Making Test B, verbal fluency (animals), and symbol digit modalities test, were used as measures of executive/attention performance. A cognitive composite, modified PACC-5, was defined as the average across z-scored cognitive tests (individually standardized towards a reference population), including ADAS delayed recall word list test (counted twice to preserve the weight on memory in the original PACC (Donohue et al, 2014)), animal fluency, MMSE and TMT-A.

## Statistical analysis

The overall statistical workflow is summarized in Fig. 6. We first evaluated 10 established machine learning regressors, including Extra Trees, Random Forest, XGBoost (eXtreme Gradient Boosting), Gradient Boosting, Bagging (Bootstrap aggregating), Ada-Boost, Decision Tree, Ridge Regression, SVM (Support vector machine), and KNN (K-nearest neighbors regression), with available fluid biomarkers, cognitive and demographic data from BF2-Initial as predictors of Aβ-PET SUVR. Appropriate tuning was done for each regressor (see Appendix Table S2). After having identified the most promising regressor variant, we continued with a careful evaluation of possible predictors of Aβ-PET in BF2-Initial. The details of model development and feature selection are elaborated in Results - Model development and feature engineering.

With the established model structure and selected feature combination, our next goal was to train an optimal model that delivers the most accurate predictions. The parameter fine-tuning of the model was first done in a 5-fold cross validation on a training set (80% of the BF2-P-MS). After that, the optimal model was trained on the whole training set and tested on a test portion (20%) with bootstrapping in 100 iterations (80 samples were randomly drawn with replacement for each iteration). We also evaluated the model performance in the independent cohort BioFINDER-1 (not at all used for training) with z-score. The optimization and tuning process of all the models mentioned in this paper was primarily guided by Bayesian search, which is a strategy for optimizing objective functions, employing a probabilistic model to balance the exploration of new parameters with the exploitation of known good parameters (Snoek et al, 2012; Pelikan et al, 1999).

While investigating feature importance of the regression models with different model architectures, we noticed fluctuations of the

results where CSF Aβ42/Aβ40 was considered the most influential biomarker derived from the Extra Trees Regressor, while plasma % P-tau217 was the most influential biomarker derived from the Gradient Boosting Regressor (see Appendix Fig. S5). We hypothesized that the fluctuations were influenced by the inclusion of Aβ-negative participants. Given this influence, together with the previous finding that some CSF and plasma biomarkers have attenuated effects within the Aβ-negative and Aβ-positive groups compared to the full range of Aβ-PET (Wisch et al, 2023), we also explored a two-step strategy that combined an initial classification of Aβ-status, with a secondary regression to quantify Aβ-load. In this strategy, instead of training regression (for continuous predictions) on the whole Aβ range, the original data was split based on the Aβ-PET SUVR (threshold 1.03 SUVR) and regressors were trained separately for Aβ-negative and Aβ-positive subjects. Given the significant changes in data distribution for these two regressors in the two-step strategy, we performed an additional round of model selection to ensure its appropriateness (see Appendix Table S4). Additionally, we used BioFINDER-1 to validate the regressor trained on the Aβ-PET data within the positive range.

In the additional results, we applied the same pipeline, used for training of the optimal models and generating the main results, to the other sub-cohorts in BioFINDER-2. Sub-cohorts from BioFINDER-1 were again utilized to validate the corresponding models (BF1-P-IA was used for the model trained on BF2-P-IA, while BF1-C-IA was used for the model trained on BF2-C-IA or BF2-C-MS)

As explained in the introduction, our a priori hypothesis was that measures of P-tau217 (in both CSF and plasma) and Aβ42/Aβ40 (especially in CSF) would be particularly influential in the

### The paper explained

**Problem**

Alzheimer's disease is characterized by the build-up of amyloid-β plaques in the brain. Measuring this pathology with positron emission tomography (PET) is accurate but expensive and not widely accessible. Cerebrospinal fluid (CSF) and blood-based biomarkers are more scalable alternatives, but it is unclear how well they can predict the actual amyloid burden measured by PET.

**The results**

Using the Swedish BioFINDER-2 cohort ($N = 1140$), we developed machine-learning models combining CSF biomarkers, plasma biomarkers and clinical variables to predict quantitative Aβ-PET results. The best models achieved $R^2 = 0.79$ and 7% mean error. The most important predictors were CSF Aβ42/Aβ40 and plasma phosphorylated-tau217 (P-tau217). CSF Aβ42/Aβ40 best identified whether amyloid plaques were present or not, while plasma P-tau217 was especially useful for estimating the degree of plaque build-up once positivity was reached.

**The impact**

Our findings indicate that soluble Aβ changes are closely linked to the presence of amyloid plaques, while plasma P-tau217 continues to increase as plaque burden builds. This highlights distinct biomarker dynamics across stages of Alzheimer's pathology, improving our understanding of disease progression and guiding how biomarkers can be applied in clinical and research settings.

models. We therefore specifically evaluated other biomarker candidates, as well as readily available demographic and cognitive features, in the presence of P-tau217 and Aβ42/Aβ40. We assessed overall model performance by the coefficient of determination ($R^2$, the proportion of the variance for the dependent variable explained by the independent variables) and Mean Absolute Percentage Error (MAPE; the absolute difference between the actual and the predicted values, divided by the actual value, averaged over the dataset).

All statistical analyses were done using Visual Studio in Python. The python libraries used mainly include pandas, NumPy, Scikit-learn, Matplotlib, Seaborn and SHAP. The representative code is available on GitHub with the repository link: https://github.com/DeMONLab-BioFINDER/continuous-abpet-prediction.

## Data availability

By request, anonymized data can be shared to qualified academic investigators. Data transfer is required to be in agreement with EU legislation on the general data protection regulation and decisions by the Ethical Review Board of Sweden and Region Skåne, which should be regulated in a material transfer agreement.

The source data of this paper are collected in the following database record: biostudies:S-SCDT-10_1038-S44321-025-00348-7.

## Peer review information

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

## Acknowledgements

Work at the authors' research center was supported by European Research Council (ADG-101096455), Alzheimer's Association (ZEN24-1069572, SG-23-1061717, ALZSI-26-1523522), GHR Foundation, Swedish Research Council (2022-00775, 2021-02219, 2025-02319), ERA PerMed (ERAPERMED2021-184), Knut and Alice Wallenberg foundation (2022-0231), Strategic Research Area MultiPark (Multidisciplinary Research in Parkinson's disease) at Lund University, Swedish Alzheimer Foundation (AF-980907, AF-994229, AF-1011799), Swedish Brain Foundation (FO2021-0293, FO2023-0163, FO2025-0055), Parkinson foundation of Sweden (1412/22), Familjen Rönnströms Stiftelse, WASP and DDLS Joint call for research projects (WASP/DDLS22-066), Michael J Fox Foundation (MJFF-025507), Cure Alzheimer's fund, Konung Gustaf V:s och Drottning Victorias Frimurarestiftelse, Skåne University Hospital Foundation (2020-O000028), Global Research and Imaging Platform, Regionalt Forskningsstöd (2022-1259) and Swedish federal government under the ALF agreement (2022-Projekt0080, 2022-Projekt0107). The precursor of $^{18}$F-flutemetamol was sponsored by GE Healthcare. KB is supported by the Swedish Research Council (#2017-00915 and #2022-00732), the Swedish Alzheimer Foundation (#AF-930351, #AF-939721, #AF-968270, and #AF-994551), Hjärnfonden, Sweden (#FO2017-0243 and #ALZ2022-0006), the Swedish state under the agreement between the Swedish government and the County Councils, the ALF-agreement (#ALFGBG-715986 and #ALFGBG-965240), the European Union Joint Program for Neurodegenerative Disorders (JPND2019-466-236), the Alzheimer's Association 2021 Zenith Award (ZEN-21-848495), the Alzheimer's Association 2022-2025 Grant (SG-23-1038904 QC), La Fondation Recherche Alzheimer (FRA), Paris, France, the Kirsten and Freddy Johansen Foundation, Copenhagen, Denmark, and Familjen Rönströms Stiftelse, Stockholm, Sweden. SES's effort is supported by National Institute on Aging R01AG070941.

## Author contributions

Niklas Mattsson-Carlgren: Conceptualization; Supervision; Funding acquisition; Methodology; Writing—original draft. Linda Karlsson: Formal analysis; Visualization; Methodology; Writing—review and editing. Weizhong Tang: Formal analysis; Investigation; Visualization; Writing—original draft. Kaj Blennow: Methodology; Writing—review and editing. Henrik Zetterberg: Methodology; Writing—review and editing. Randall J Bateman: Methodology; Writing—review and editing. Suzanne E Schindler: Methodology; Writing—review and editing. Nicolas Barthelemy: Methodology; Writing—review and editing. Sebastian Palmqvist: Funding acquisition; Investigation; Methodology; Writing—review and editing. Erik Stomrud: Investigation; Project administration; Writing—review and editing. Shorena Janelidze: Investigation; Methodology; Writing—review and editing. Oskar Hansson: Conceptualization; Supervision; Funding acquisition; Writing—review and editing.

Source data underlying figure panels in this paper may have individual authorship assigned. Where available, figure panel/source data authorship is listed in the following database record: biostudies:S-SCDT-10_1038-S44321-025-00348-7.

## Funding

## Disclosure and competing interests statement

NMC has received consulting/speaker fees from BioArctic, Biogen, Eli Lilly, Merck, Novo Nordisk, and Owkin. KB has served as a consultant and at advisory boards for Abbvie, AC Immune, ALZPath, AriBio, BioArctic, Biogen, Eisai, Lilly, Moleac Pte. Ltd, Neurimmune, Novartis, Ono Pharma, Prothena, Roche Diagnostics, and Siemens Healthineers; has served at data monitoring committees for Julius Clinical and Novartis; has given lectures, produced educational materials and participated in educational programs for AC Immune, Biogen, Celdara Medical, Eisai and Roche Diagnostics; and is a co-founder of Brain Biomarker Solutions in Gothenburg AB (BBS), which is a part of the GU Ventures Incubator Program, outside the work presented in this paper. SP has acquired research support (for the institution) from ki elements/ADDF and Avid. In the past 2 years, he has received consultancy/speaker fees from Bioartic, Biogen, Esai, Eli Lilly, and Roche. SES has served on scientific advisory boards on biomarker testing and clinical treatment pathways for Eisai and Novo Nordisk, and has received speaking fees from Eli Lilly. OH is an employee of Lund University and Eli Lilly.

# Expanded View Figures

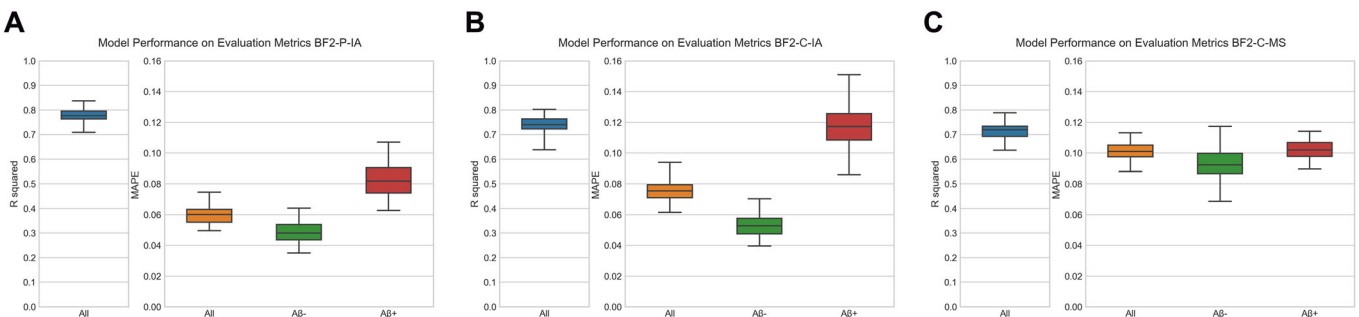

**Figure EV1. Performance of models trained on sub-cohorts.**

Panels (**A–C**) illustrate the bootstrapping evaluation results on the test sets for the sub-cohorts BF2-P-IA (**A**), BF2-C-IA (**B**), and BF2-C-MS (**C**), with $R^2$ and MAPE on the whole Aβ-PET range, and the metrics MAPE_POS and MAPE_NEG on the Aβ-positive and Aβ-negative range, respectively. The box plots show the quartiles of these 100 iterations (IQR = Q3–Q1), with median as center line, and whiskers extending to minimum and maximum points.

