## [Peer Review File · EMBO Molecular Medicine]

Prediction of continuous amyloid PET with fluid measures of phosphorylated tau and beta-amyloid

Niklas Mattsson-Carlsson, Linda Karlsson, Weizhong Tang, Kaj Blennow, Henrik Zetterberg, Randall Bateman, Suzanne Schindler, Nicolas Barthelemy, Sebastian Palmqvist, Erik Stomrud, Shorena Janelidze, and Oskar Hansson

Corresponding authors: Niklas Mattsson-Carlsson (niklas.mattsson-carlsson@med.lu.se) , Oskar Hansson (oskar.hansson@med.lu.se)

Review Timeline:

Submission Date:	18th Jun 25
Editorial Decision:	12th Aug 25
Revision Received:	28th Oct 25
Editorial Decision:	5th Nov 25
Revision Received:	7th Nov 25
Accepted:	13th Nov 25

Editor: Jingyi Hou

Transaction Report:

12th Aug 2025

Dear Dr. Mattsson-Carlgrén,

Thank you again for submitting your work to EMBO Molecular Medicine. We have now received the reports from the two reviewers and as you will see below, the reviewers think that the study is interesting. The reviewers have raised several mostly minor concerns, which we ask you to address in a revision.

I think the referees' recommendations are clear and need not be repeated here. All issues raised by the referees need to be satisfactorily addressed. As you may already know, our editorial policy allows in principle a single round of major revision so it is essential to provide responses to the referees' comments that are as complete as possible. Please feel free to contact me in case you would like to discuss in further detail any of the issues raised by the referees.

Please also contact us as soon as possible if similar work is published elsewhere. If other work is published, we may not be able to extend the revision period beyond three months.

I look forward to receiving your revised manuscript soon.

Use this link to login to the manuscript system and submit your revision: <https://embomolmed.msubmit.net/cgi-bin/main.plex>

Sincerely,
Jingyi

Jingyi Hou
Senior Editor
EMBO Molecular Medicine

We require:

- 1) A .docx formatted version of the manuscript text (including legends for main figures, EV figures and tables). Please make sure that the changes are highlighted to be clearly visible.
- 2) Individual production quality figure files as .eps, .tif, .jpg (one file per figure). For guidance, download the 'Figure Guide PDF': (<https://www.embopress.org/page/journal/17574684/authorguide#figureformat>).
- 3) A .docx formatted letter INCLUDING the reviewers' reports and your detailed point-by-point responses to their comments. As part of the EMBO Press transparent editorial process, the point-by-point response is part of the Review Process File (RPF), which will be published alongside your paper.
- 4) A complete author checklist, which you can download from our author guidelines (<https://www.embopress.org/page/journal/17574684/authorguide#submissionofrevisions>). Please insert information in the checklist that is also reflected in the manuscript. The completed author checklist will also be part of the RPF.
- 5) Please note that all corresponding authors are required to supply an ORCID ID for their name upon submission of a revised manuscript.

6) It is mandatory to include a 'Data Availability' section after the Materials and Methods. Before submitting your revision, primary datasets produced in this study need to be deposited in an appropriate public database, and the accession numbers and database listed under 'Data Availability'. Please remember to provide a reviewer password if the datasets are not yet public (see <https://www.embopress.org/page/journal/17574684/authorguide#dataavailability>).

12) Author contributions: You will be asked to provide CRediT (Contributor Role Taxonomy) terms in the submission system. These replace a narrative author contribution section in the manuscript.

13) A Conflict of Interest statement should be provided in the main text.

14) Please provide a 'Synopsis' to further enhance discoverability. Synopses are displayed on the journal webpage and are freely accessible to all readers. They include a short stand first (maximum of 300 characters, including space) as well as 2-5 one-sentences bullet points that summarizes the paper. Please write the bullet points to summarize the key NEW findings. They should be designed to be complementary to the abstract - i.e. not repeat the same text. We encourage inclusion of key

acronyms and quantitative information (maximum of 30 words / bullet point). Please use the passive voice. Please attach these in a separate file or send them by email, we will incorporate them accordingly.

Please also suggest a visual abstract to illustrate your article as a PNG file 550 px wide x 300-600 px high.

15) All Materials and Methods need to be described in the main text using our 'Structured Methods' format. According to this format, the Methods section includes a Reagents and Tools Table (listing key reagents, experimental models, software and relevant equipment and including their sources and relevant identifiers) followed by a Methods and Protocols section describing the methods, ideally using a step-by-step protocol format. The aim is to facilitate adoption of the methodologies across labs.

Please download and fill our Reagents and Tools Table template (.docx), which you can find in our author guidelines: <https://www.embopress.org/page/journal/17574684/authorguide#structuredmethods>

***** Reviewer's comments *****

Referee #1 (Remarks for Author):

It is a great pleasure to participate in the review of this article. Overall, the author's approach is quite interesting. The integration of machine learning to explore optimal blood biomarkers for Alzheimer's disease makes this study distinctive. However, the following points require clarification:

The characterization of the cohort in the Results section is confusing. The remarkably low proportion of cognitively impaired participants yet a 37.1% PET-positive rate raises questions. Furthermore, the author did not clearly explain why discrepancies arose when processing data from different subpopulations.

While the presented data suggest the model is robust, I am curious about the origin of these 10 different regressor methods. Were algorithms with potentially higher predictive performance considered or evaluated?

Referee #2 (Comments on Novelty/Model System for Author):

In this manuscript, the authors use the BioFINDER-1 and -2 studies to develop machine learning algorithms to predict AB-PET using plasma and/or CSF biomarker measurements. This is an important question as understanding AB status can aid in diagnosis, prognosis, and in potential clinical trials. Furthermore, the availability of biomarkers are at lower cost and invasiveness, and could be broadly applicable where PET is not available. The study is unique in that it attempts to model not just AB positive vs negative, but also the continuous measures of AB-PET uptake. This uses a well established cohort and cutting edge technologies and analyses to address a potential important question.

Referee #2 (Remarks for Author):

Overall, the manuscript addresses an interesting and important question in the field of biomarker analysis for dementia diagnosis and monitoring. The manuscript is generally clearly written and the experimental design, results, and conclusions are well supported. I think this warrants publication, though they are a few minor revision suggestions I would recommend.

1 - the Introduction is quite densely written. A division of thoughts instead of two very long paragraphs could make for ease of reading.

2 - in the final feature selection it is stated that cognitive status/results added only minor increases. However, the number of individuals with dementia is very low in the dataset. Moreover, the number of APOE4/4 carriers is also quite low. I think that this should be addressed in the limitation section.

3 - again as a limitation to address, the introduction mentions using biomarker in a global context where AB-PET is unavailable. However, the genetic background and environmental context of the samples used in this training is quite homogenous. A statement regarding how this might impact global populations should be added.

Reviewer's comments

Referee #1 (Remarks for Author):

It is a great pleasure to participate in the review of this article. Overall, the author's approach is quite interesting. The integration of machine learning to explore optimal blood biomarkers for Alzheimer's disease makes this study distinctive. However, the following points require clarification:

The characterization of the cohort in the Results section is confusing. The remarkably low proportion of cognitively impaired participants yet a 37.1% PET-positive rate raises questions. Furthermore, the author did not clearly explain why discrepancies arose when processing data from different subpopulations.

Authors' response: We thank the reviewer for this important comment and are happy to clarify. In BioFINDER-2, A β -PET was generally not performed in participants with dementia, by study design. This explains the relatively low proportion of cognitively impaired participants in our sample. Moreover, the cognitively unimpaired study arms of BioFINDER-2 were deliberately enriched for APOE ϵ 4 carriers, which further increases the expected prevalence of A β -positivity. Taken together with global prevalence estimates of A β -positivity (according to Jansen et al. (2022), the prevalence of A β -positivity is approximately 20% in cognitively unimpaired individuals around age 65 (average age of our sample), and ~45% in individuals with MCI), these factors explain the observed A β -positivity rate.

For clarity, we have now stated as a limitation that A β -PET was generally not performed in participants with dementia (page 9, lines 341ff):

“A fourth limitation was that, according to the BF2 study protocol, A β -PET was rarely performed in individuals with dementia, which limited our ability to evaluate model performance at this disease stage.”

Additionally, we also now comment on the A β -positivity prevalence in our sample (page 4, lines 73-75):

“The overall rate of A β -PET positivity was 37.1% (consistent with that the cognitively unimpaired study arms of BioFINDER-2 were enriched for APOE ϵ 4 carriers and with established global prevalence estimates).”

With regards to different subpopulations, we found relatively robust results across populations (but somewhat lower for CSF based tests for p-tau217 compared to plasma p-tau217), as described in the sensitivity analysis section (page 7, lines 223-225): “Specifically, the mean R² scores averaged around 0.74 and 0.72 for BF2-C-IA and BF2-C-MS respectively, in contrast to the mean R² scores of 0.83 and 0.78 for BF2-P-MS and BF2-P-IA.”

While the presented data suggest the model is robust, I am curious about the origin of these 10 different regressor methods. Were algorithms with potentially higher predictive performance considered or evaluated?

Authors response: We appreciate the opportunity to expand on the description and discussion of the ML models. We recognize that there are many possible ML algorithms with

varying levels of complexity and predictive performance. Our goal in this study was to sample broadly across different classes of models, linear (e.g., linear regression, ridge), tree-based (e.g., random forest, gradient boosting, extra trees), and kernel-based (e.g., support vector regression), to capture a wide range of possible relationships within the data. We focused on algorithms that are well established, widely used, and known to perform robustly across diverse regression tasks.

In addition, we systematically explored different input feature combinations, performed hyperparameter tuning, and cross-validation procedures to ensure a rigorous evaluation. While it is possible that other methods (e.g., deep learning architectures) might achieve higher predictive performance, we aimed to balance model diversity with computational feasibility. Within this framework, the Extra Trees Regressor achieved the best performance. We agree that future work could extend this search to include more complex or emerging algorithms, which we now have noted in the Discussion/Limitations (page 9, lines 337ff):

“Third, while we focused on a range of well-established models and rigorously optimized their features and hyperparameters, future work could explore more advanced algorithms, such as deep learning architectures (which may need larger sample sizes for efficient training), to potentially further improve predictive performance.”

Referee #2 (Comments on Novelty/Model System for Author):

In this manuscript, the authors use the BioFINDER-1 and -2 studies to develop machine learning algorithms to predict AB-PET using plasma and/or CSF biomarker measurements. This is an important question as understanding AB status can aid in diagnosis, prognosis, and in potential clinical trials. Furthermore, the availability of biomarkers are at lower cost and invasiveness, and could be broadly applicable where PET is not available. The study is unique in that it attempts to model not just AB positive vs negative, but also the continuous measures of AB-PET uptake. This uses a well established cohort and cutting edge technologies and analyses to address a potential important question.

Referee #2 (Remarks for Author):

Overall, the manuscript addresses an interesting and important question in the field of biomarker analysis for dementia diagnosis and monitoring. The manuscript is generally clearly written and the experimental design, results, and conclusions are well supported. I think this warrants publication, though they are a few minor revision suggestions I would recommend.

1 - the Introduction is quite densely written. A division of thoughts instead of two very long paragraphs could make for ease of reading.

Authors' response: We thank the reviewer for this suggestion and agree that the introduction would benefit from such a division. We have therefore now separated the introduction into five paragraphs instead of two. These paragraphs focus on 1) novel AD treatments increasing the need for methods to quantify A β burden, 2) existing PET, CSF and plasma biomarkers related to A β , 3) previous studies using fluid biomarkers to predict A β PET status, 4) previous efforts to predict continuous A β PET burden, and 5) our specific aims and hypotheses.

2 - in the final feature selection it is stated that cognitive status/results added only minor increases. However, the number of individuals with dementia is very low in the dataset. Moreover, the number of APOE4/4 carriers is also quite low. I think that this should be addressed in the limitation section.

Authors' response: We thank the reviewer for this valuable observation and fully agree that it is important to address. The number of individuals with dementia in our dataset is low, which is due to that in the BioFINDER-2 study protocol, amyloid PET was generally not performed in participants with dementia. Consequently, cognitive status may have contributed more strongly if a larger number of dementia cases had been included. The lower number of dementia patients may also be expected to somewhat reduce the number of APOE ϵ 4 homozygotes. In addition, inclusion in BioFINDER-2 was stratified by APOE ϵ 4 status among cognitively unimpaired individuals. Together, this may influence the apparent contribution of APOE ϵ 4 compared to what would be expected in a population with frequencies closer to global prevalence. We agree that both of these points should be explicitly acknowledged in the Discussion/Limitations section, and we have revised the manuscript accordingly (page 9, lines 341ff):

“A fourth limitation was that, according to the BF2 study protocol, A β -PET was rarely performed in individuals with dementia, which limited our ability to evaluate model performance at this disease stage. Certain features, such as cognitive status, may have contributed more strongly to prediction if a larger number of dementia cases had been included. In addition, inclusion in BF2 was stratified by APOE ϵ 4 status among cognitively unimpaired individuals, which may have influenced the contribution of this feature compared to a population with frequencies closer to global prevalence.”

3 - again as a limitation to address, the introduction mentions using biomarker in a global context where AB-PET is unavailable. However, the genetic background and environmental context of the samples used in this training is quite homogenous. A statement regarding how this might impact global populations should be added.

Authors' response: We thank the reviewer for highlighting this important point. We agree that the relatively homogenous genetic and environmental background of the BioFINDER-2 cohort could limit the direct generalizability of our findings to more diverse global populations. We have now added a statement in the Discussion/Limitations section to explicitly note this and emphasize the need for validation in more diverse populations (page 9, lines 347ff):

“Another limitation was the relatively homogenous genetic and environmental background of BF2, factors that could potentially influence biomarker performance. Even though the models generalized well to the independent cohort BF1, findings should be validated across more diverse populations in future studies.”

5th Nov 2025

Dear Dr. Mattsson-Carlgrén,

Thank you for the submission of your revised manuscript to EMBO Molecular Medicine. We have now received the enclosed report from the referee who was asked to re-assess it. As you will see, the referee is now supportive, and I am pleased to inform you that we will be able to accept your manuscript pending the following amendments:

1. Please remove the figures from the manuscript file and upload them as separate, high-resolution figure files. The legends should stay in the manuscript text. Please add the heading "Expanded View Figure Legends" after the main tables and before the EV figure legends.

2. Please add missing callouts for panels A,B of Figre 5 and panels A,B,C of Figure EV1.

3. "Funding" section needs to be renamed to "Acknowledgments". Please verify that the funding details (including project numbers) in the manuscript match those entered in the submission system. The line-by-line entries provided in our system will be automatically linked upon publication, so accuracy and completeness are essential.

4. Appendix: please add page numbers in the Table of Contents.

5. Please move the information in "Ethics" to the "Methods" section where appropriate.

6. Source data: please upload then as one (Zip) file per figure.

7. Data availability : Please revise the following sentence "By request, anonymized data can be shared to qualified academic investigators for the purpose of replicating procedures and results presented in the article."

to:

"By request, anonymized data will be shared to qualified academic investigators."

8. Please download and fill our Reagents and Tools Table template (.docx), which you can find in our author guidelines: <https://www.embopress.org/page/journal/17574684/authorguide#structuredmethods>

9. "Disclosure" should be renamed to "Disclosure and Competing Interests Statement".

10. Please move the "summary" to the manuscript file and change the heading to "The Paper Explained".

11. Please upload the EV tables as two separate files and remove them from the manuscript text.

12. The references need to be formatted according to the EMBO Molecular Medicine reference style. Citations should be listed in alphabetical order. Please list up to 10 co-authors of a paper before adding et al. in the reference list. Remove the DOIs for all published articles.

13. Please correct the order of the manuscript sections to: Abstract / Keywords / The Paper Explained / Introduction /Results / Discussion / Methods / Data Availability /Acknowledgements / Disclosure and Competing Interests Statement / References / Figure Legends / Tables / Expanded View Figure Legends

14. Please note that the box plots need to be defined in terms of minima, maxima, centre, bounds of box and whiskers, and percentile in the legends of figures 2C, 4, 5, EV1 A-C.

Please submit your revised manuscript within two weeks.

I look forward to reading a new revised version of your manuscript as soon as possible.

Kind regards,
Jingyi

Jingyi Hou
Senior Editor
EMBO Molecular Medicine

*** Instructions to submit your revised manuscript ***

***** Reviewer's comments *****

Referee #1 (Comments on Novelty/Model System for Author):

This study presents an innovative machine learning approach using BioFINDER-1 and -2 data to predict A β -PET from plasma and CSF biomarkers. The work is methodologically sound, clinically meaningful, and clearly presented.

Referee #1 (Remarks for Author):

The authors have responded carefully and effectively to all reviewer comments. The revised manuscript is much improved and suitable for consideration.

The authors addressed the remaining editorial issues.

13th Nov 2025

Dear Dr. Mattsson-Carlgrén,

We are pleased to inform you that your manuscript is accepted for publication and is now being sent to our publisher to be included in the next available issue of EMBO Molecular Medicine.

Yours sincerely,
Jingyi

Jingyi Hou
Senior Editor
EMBO Molecular Medicine
